# How Does the Gus Schumacher Nutrition Incentive Program Work? A Theory of Change

**DOI:** 10.3390/nu14102018

**Published:** 2022-05-11

**Authors:** Kirsten H. Leng, Amy L. Yaroch, Nadine Budd Nugent, Sarah A. Stotz, James Krieger

**Affiliations:** 1Healthy Food America, Seattle, WA 98122, USA; jkrieger@hfamerica.org; 2Gretchen Swanson Center for Nutrition, Omaha, NE 68154, USA; ayaroch@centerfornutrition.org (A.L.Y.); dnugent@centerfornutrition.org (N.B.N.); 3Centers for American Indian and Alaska Native Health, Colorado School of Public Health, University of Colorado Anschutz Medical Campus, Aurora, CO 80045, USA; sarah.stotz@cuanschutz.edu; 4School of Public Health, University of Washington, Seattle, WA 98195, USA

**Keywords:** nutrition incentives, fruit and vegetables access/intake, health equity, food security, low income, food systems, theory of change, Supplemental Nutrition Assistance Program (SNAP), Gus Schumacher Nutrition Incentive Program (GusNIP)

## Abstract

Increased fruit and vegetable (FV) intake is associated with decreased risk of nutrition-related chronic diseases. Sociodemographic disparities in FV intake indicate the need for strategies that promote equitable access to FVs. The United States Department of Agriculture’s Gus Schumacher Nutrition Incentive Program (GusNIP) supports state and local programs that offer nutrition incentives (NIs) that subsidize purchase of FVs for people participating in the Supplemental Nutrition Assistance Program (SNAP). While a growing body of research indicates NIs are effective, the pathways through which GusNIP achieves its results have not been adequately described. We used an equity-focused, participatory process to develop a retrospective Theory of Change (TOC) to address this gap. We reviewed key program documents; conducted a targeted NI literature review; and engaged GusNIP partners, practitioners, and participants through interviews, workshops, and focus groups in TOC development. The resulting TOC describes how GusNIP achieves its long-term outcomes of increased participant FV purchases and intake and food security and community economic benefits. GusNIP provides NIs and promotes their use, helps local food retailers develop the capacity to sell FVs and accept NIs in accessible and welcoming venues, and supports local farmers to supply FVs to food retailers. The TOC is a framework for understanding how GusNIP works and a tool for improving and expanding the program.

## 1. Introduction

Fruit and vegetable (FV) intake is associated with decreased risk of chronic diseases [1,2]. Yet, Americans generally do not eat enough FVs to meet federal recommendations [3]. Fruits and vegetables (FVs) are often more expensive than non-perishable, less healthful ‘convenience’ foods [4]. Many households with low incomes cannot afford to purchase necessary quantities and varieties of FVs needed to meet dietary guidelines [4,5,6,7]. The accessibility, availability, and affordability of FVs are influenced by the structural environment of economically distressed communities and shaped by historical and political contexts [8].

To promote increased FV intake among people with low incomes, Congress established the Gus Schumacher Nutrition Incentive Program (GusNIP) in 2018 [9,10,11], building on the preceding Food Insecurity Nutrition Incentive (FINI) program [12]. GusNIP is authorized to spend USD 250 million over five years and is administered by the United States Department of Agriculture (USDA). The largest component of GusNIP is a competitive grant program that funds state and local non-profit organizations to provide financial nutrition incentives (NIs) that subsidize purchases of FVs to people who participate in the Supplemental Nutrition Assistance Program (SNAP). There are currently 65 funded NI grantee projects (henceforth called “projects”) [13].

People who receive NIs use them at the point of sale to purchase fresh, canned, dried, or frozen FVs which do not contain added sugars, fats/oils, or salt. For each SNAP dollar spent on eligible FVs, participants receive a matching amount of NIs [14]. They can use NIs at participating farm direct (e.g., farmers markets, mobile markets) and brick and mortar (e.g., supermarkets, small grocery stores) retailers (henceforth called “food retailers”). Program designs vary across grantees with respect to types of FVs included, match amounts, retailer types, and populations served to better meet local conditions and preferences.

The USDA also funds the GusNIP Training, Technical Assistance, Evaluation, and Information Center (NTAE), which provides training, technical assistance, and reporting and evaluation guidance to grantees [15]. The NTAE created the Nutrition Incentive Hub, a coalition of partners and scientific advisors, which provides consultations, resources, networking opportunities, and best practices to GusNIP grantees and practitioners [16].

Well-designed studies have demonstrated that NIs are associated with moderate increases in FV intake [17,18,19,20,21]. Modelling studies suggest NIs may be cost-effective relative to accepted thresholds for cost per quality-adjusted life years gained due to long-term reductions in chronic disease incidence that yield substantial savings in health care costs [22,23]. However, a deeper understanding of how GusNIP achieves its outcomes is needed to guide development of best practices and foster a deeper shared understanding of the program across grantees, future applicants, funders, evaluators, and advocates. A Theory of Change (TOC) is a tool for understanding how and why a program works [24,25,26,27,28,29]. It identifies what a program hopes to achieve and causal pathways leading to its outcomes. It describes the environment in which a program operates and assumptions about what is needed for success. A TOC is developed using a structured and participatory process that includes a synthesis of current evidence, review of program documents, and interactive input from people implementing and benefitting from the program and from partners with expertise in program design, funding, and evaluation. A TOC is a living theory which is refined as new evidence emerges, the environment changes, and experience is gained [24,25,26,27,28,30].

We are not aware of any published TOCs for NI programs; therefore, we developed a TOC for GusNIP to fill this gap. This paper describes the TOC, the equity-centered process used to create it, and lessons learned from its development.

## 2. Materials and Methods

The NTAE and two expert consultants (henceforth called “facilitators”) collaboratively organized the TOC development process. The NTAE commissioned preparation of the TOC, contributed knowledge of the overarching GusNIP program and grantee projects, and participated in TOC development. The facilitators were knowledgeable about NIs, TOC methods, and equity-centered processes. They managed the TOC development process and drafted the TOC.

TOC contributors included the NTAE, GusNIP Nutrition Incentive Hub partners (henceforth called “partners”), people implementing NI projects (henceforth called “practitioners”), and people using NIs (henceforth called “participants”) (Table 1).

TOC development occurred a year after the GusNIP program launch, between September 2020 and December 2021. The TOC development process followed best practices and included standard TOC elements (Table 2) [26,30].

Facilitators employed an equity lens to foster an equitable approach to the TOC development process and incorporate equity considerations in the TOC itself [32]. The definition of equity used - the condition that would be achieved if one’s social status, including race, income and wealth, and place of residence no longer influenced how one fares - was derived from Racial Equity Tools [33].

The elements of the multi-step TOC process we implemented are described in detail below and in Figure 1. They included GusNIP program documents; published evidence; and findings from interviews, workshops, and focus groups we conducted to develop the TOC. Additional details are available in Appendix B.

### 2.1. Literature and Document Reviews

The facilitators reviewed the GusNIP 2021 Request for Applications [14], the 2018 GusNIP Farm Bill statute [34], internal NTAE documents, and grant reporting documents. They extracted information to generate initial descriptions of TOC elements and inform interview questions and workshop agendas. They reviewed literature extracted from two systematic reviews underway or completed by the authors at the inception of the TOC development process to identify current knowledge about NIs [35,36]. The first review searched PubMed to identify healthy food pricing incentive studies published between 2000 and 2018. Additional articles were found by searching the Cochrane Library and Google Scholar using a limited set of keywords, asking experts in the field, and scanning reference lists of relevant articles. The review was limited to peer-reviewed, full-length, English-language articles with original data.

The second review identified relevant literature in PubMed, the National Institute of Food and Agriculture (NIFA) Current Research Information Systems (CRIS), and Google Scholar. Peer-reviewed manuscripts, project summaries, reports, theses, and abstracts were included between the years 2009 and 2019.

PubMed was selected as a comprehensive source of peer-reviewed biomedical journal with relevant publications on NI literature. Google Scholar was used to identify relevant grey literature sources. The Cochrane Library was used to identify systematic reviews. NIFA CRIS is a federal government repository of all current and past FINI or GusNIP NI projects [37]. NI project websites listed in the NIFA CRIS repository were searched and any reports or literature relevant to NI projects on individual websites were identified and reviewed.

To update these reviews, the facilitators searched PubMed and Google Scholar for articles describing any type of healthy food pricing incentive strategy published between 1 January 2018 and 2 July 2021. They screened titles and abstracts to determine potential articles for inclusion. Next, they conducted full text reviews to identify articles for data extraction (protocol available in Appendix A). They found 57 articles describing NI projects, processes, and outcomes relevant to the TOC across all sources. The evidence review informed development of the initial TOC and served as an evidence base for the final TOC.

### 2.2. Human Subjects Protection

The interviews and focus groups were approved by the University of Nebraska Medical Center IRB. TOC contributors gave implied consent for participation by agreeing to participate with the understanding that their statements were confidential.

### 2.3. Interviews

One facilitator interviewed partners and practitioners (*n* = 24) representing 18 organizations to introduce them to the TOC concept and identify TOC elements from their perspectives. The facilitator used a semi-structured interview guide (see Appendix A) based on a backwards mapping process which first identified the program’s ultimate goals and then worked “backwards” towards the earliest outcomes needed to reach the goals [24,29]. Interviews took place on the Zoom [38] platform, lasted 60–90 min, and were recorded with permission. The facilitator took notes on a shared screen during each interview to allow interviewees to build and edit TOC pathways in real time. The facilitator probed about GusNIP’s intended and unintended equity impacts.

Additionally, the facilitator interviewed two participants to learn about their experiences with GusNIP. The 45–60 min interviews were conducted via telephone using a semi-structured interview guide (see Appendix A) and documented with detailed notes. Interviewees received a $50 gift card stipend.

### 2.4. Workshops 

The NTAE and facilitators hosted three 2–2.5 h workshops offered via Zoom with partners and practitioners to discuss and refine the initial TOC. Workshops were recorded (with attendee permission) and documented by notetakers. Workshop formats included presentations, small breakout group discussions (5–7 attendees per group), and whole-group brainstorming and consensus building sessions. Honoraria ($250 per workshop) were provided to those who participated in workshops outside their core job requirements. The first two workshops oriented attendees to the TOC process and discussed TOC goals, outcomes, and pathways. The third focused on equity, assumptions, and environmental context. After the third workshop, attendees were invited to provide additional written input.

### 2.5. Focus Groups

Facilitators conducted three focus groups to ground the initial TOC in participant lived experiences. NI projects in Tennessee, Pennsylvania, and California purposively recruited people who participated in NI projects (*n* = 17) to include those who shopped at farmers markets and grocery stores, and who preferred speaking in English and Spanish. Participants received a $75 gift card incentive. A Spanish-speaking partner conducted the California group in Spanish. A Spanish-English translator provided real time translation for recording purposes. Focus groups were conducted over Zoom and were recorded with participant permission. Facilitators noted key points and non-verbal cues indicating participant agreement or disagreement with discussion points. Audio recordings were transcribed using a professional transcription service [39]. Facilitators used thematic analysis to organize data by the sequential steps participants take to use NIs and by TOC component [40].

### 2.6. TOC Drafting

The development of the TOC was iterative. The facilitators triangulated data from the program document and literature reviews and interviews to develop the initial TOC, using thematic analysis to organize data by TOC component [40,41]. TOC modifications captured from workshop and focus group discussions were applied to the draft. The facilitators then solicited additional information from TOC contributors as needed and prepared a simplified, final draft version of the TOC to distribute to partners and practitioners for review. Facilitators incorporated this feedback into a final TOC with guidance from the NTAE. The NTAE and the facilitators met regularly to resolve differences in perspectives throughout the TOC process.

## 3. Results

The following sections summarize learnings from the TOC development process and present the TOC.

### 3.1. Evidence and Perspectives from the TOC Development Process

#### 3.1.1. Equity

Equity emerged as a core value of GusNIP. By design, GusNIP addresses economic and racial disparities in FV access and intake by its focus on people enrolled in SNAP, who must have low incomes to be eligible, and are predominantly non-white [42,43].

TOC contributors used an equity lens to examine GusNIP TOC components with respect to race, ethnicity, citizenship status, language, socioeconomic status, and geography and consideration of the historical context of racism in the food system. GusNIP moves the food system towards equity by tailoring projects to the diverse participants served and engaging diverse retailers. Equity is advanced when community members play meaningful roles in project planning, implementation, and evaluation. GusNIP projects and national partners have begun efforts to share important program information and develop mechanisms to share power between national and local partners, grantees, and participants. For example, individualized annual impact reports are provided to each grantee so they can share results with local partners and communities. The work to center GusNIP-funded projects in equity is iterative and ongoing.

#### 3.1.2. Activities, Short-Term Outcomes, Pathways, and Assumptions

TOC contributors identified TOC components and built out pathways. They stressed the importance of a foundation of partnerships between the grantee and retailers, community organizations, participants, and the national partners. They validated the core assumptions that local grantee organizations and food retailers with sufficient capacity to implement NI projects exist and want to participate and that participants want to eat more FVs.

Participants validated the three pathways that lead to participants successfully using NIs. They described the importance of knowing about NIs and the role participants play in educating community members about projects. They expressed their desires to support local farmers and eat local FVs and noted that NIs make these possible. One participant shared: *“Farmers markets can be more expensive, so it is great to be able to get that good food and also be supporting more local farmers and local businesses … (with NIs).”* They emphasized that easily accessible food retailers with well-trained and welcoming staff increased their interest in using NIs.

The NI literature review revealed variation in the strength of evidence for pathways. For example, evidence describing what participants need to successfully use NIs was well-developed [18,44,45,46,47,48,49,50,51]. However, there was no evidence describing how projects can support participation by farmers. Evidence is cited in the section below describing the TOC.

#### 3.1.3. Long-Term Outcomes

Partners and practitioners noted that increased FV intake and food security among participants and expanded economic benefits for participants, food retailers, and farmers are important outcomes along with increased purchases of FVs, the primary goal in federal program documents [9,14]. Participants agreed. One shared: “*(NIs) extend your buying power and your ability to keep your nutrition at the highest level possible*”.

#### 3.1.4. Ultimate Goals

The primary goals for GusNIP identified by Congress and the USDA are “increased FV purchase among low-income consumers [9]” and bringing together “stakeholders from distinct parts of the food system [14].” TOC contributors expanded these goals to include improving community health and economic well-being and sustaining and expanding a diverse and just local/regional food system. Participants and findings from the evidence review supported these broader goals. One focus group participant spoke about NIs improving health: “*(Before the NI program) … I did not have a lot of money to buy FVs and [my kids] would get sick… The doctor would tell me, ‘Your kids are gaining weight, they need to eat more FVs…’ I would love to give my kids more FVs, but I don’t have the money for it. But now that I get the help from (NIs), I can do this*”.

Partners and practitioners discussed the trade-offs in pursuing these goals. On one hand, GusNIP strives to maximize access and affordability of FVs for people using SNAP. This suggests offering NIs at larger brick and mortar retailers where NI participants are already using their SNAP benefits and where FVs may be relatively less expensive and more consistently available year-round compared to farm direct retailers. On the other hand, GusNIP also aims to support a local food system that is resilient, robust, and supportive of a healthy environment. To work towards this goal, projects include farm direct retailers and may consider local/regional procurement requirements for brick and mortar retailers. Both goals have merit and were supported by focus group participants. Many participants valued being able to use NIs at farm direct sites. Supporting farmers and consuming local FVs was a core value for them and NIs made this possible. Other focus group participants wanted to be able to use NIs at less expensive, larger retail stores. Each project determines its focus and design by balancing these goals to best meet its community’s priorities and needs.

#### 3.1.5. Environmental Context

TOC contributors agreed that a strength of GusNIP is the flexibility to adapt local projects to the specific environmental contexts of each community. For example, projects can prioritize providing local FVs if they are readily available in their community. They can prioritize working with food retailers that provide FVs that are culturally appropriate for their community.

### 3.2. Theory of Change

The diagram in Figure 2 summarizes the TOC. The following narrative provides further description.

#### 3.2.1. Pathways, Short-Term Outcomes, and Activities

The TOC starts with a pathway describing the strong foundation of GusNIP. National partners and advisors (members of the NTAE and Nutrition Incentive Hub) and grantees partner to implement equity-centered, efficient GusNIP projects. The Nutrition Incentive Hub provides technical assistance to grantees, convenes a learning community of grantees and other NI practitioners, and facilitates project evaluation. This supports grantees to develop or enhance their capacities (e.g., skills to plan and implement NI projects, engage community members, provide required reporting), secure the necessary resources, and build or expand a robust network of partnerships with community organizations and participants needed to successfully implement their projects. Grantees are knowledgeable about their communities, inform them about projects, and engage community members and organizations in project design, implementation, and evaluation decision-making.

With this foundation in place, grantees implement activities that lead to short-term outcomes through three distinct and interconnected pathways:**Participants want FVs and participate in NI projects**. Projects develop community-centered and culturally tailored promotion strategies and local partnerships to encourage FV purchases and intake, inform the community about the availability of NIs and how to use them, and distribute NIs. Some projects provide education about nutrition and FV preparation. Projects may address barriers to FV purchasing, such as location of FV food retailers.**Food retailers are accessible, welcoming, and provide FVs to participants**. Projects help food retailers learn more about participants’ needs and the GusNIP program. They support food retailers to develop the infrastructure needed to accept NIs (e.g., procedures and technology for point-of-sale processing of NIs) and offer high quality FVs. They encourage participation by culturally diverse and locally owned and operated food retailer sites that may be the preferred shopping locations of participants and provide culturally tailored and welcoming retail environments (e.g., offer FVs preferred by participants, hire staff who speak the language(s) of the community and use signage in these language(s)). Projects work with food retailers to identify the form of NI (e.g., paper voucher, token, electronic) most appropriate for their retail setting and community. Projects may build food retailer capacity to sell FVs by connecting them to FV suppliers (including local and regional farmers, when feasible) and supporting promotional activities.**Local farmers supply FVs to some retail food stores and at farm direct sites.** Projects facilitate sales of FVs by local and culturally diverse farmers to participating food retailers and acceptance of NIs at farm direct retail sites. Some projects strengthen local FV distribution channels by requiring participating retailers to source produce locally and supporting partnerships between food retailers and local farmers.

Appendix C provides additional examples of activities that support the TOC pathways.

The TOC includes a hypothesized positive feedback loop among the participant, food retailer, and local farmer pathways that results in participants buying more FVs: NIs provide participants with added FV purchasing power, thus increasing their ability to buy FVs. Food retailers and farmers respond by increasing the supply of FVs, which can further increase FV purchases. The increased FV supply and sales may spur expansion of local food systems, which in turn can further increase FV supply. As retailers provide more quality FVs and learn how to create settings that are welcoming, culturally tailored, and accessible, more community members shop at these locations, further increasing community FV purchasing and leading to additional local economic and food system benefits.

The extent to which NIs increase FV purchases depends in part on how they are designed. Projects choose the NI amount and frequency of issuance. Incentives given for a longer duration of time increase FV purchases and intake [52,53]. However, some projects choose smaller per-household NI amounts so that more community members can participate.

#### 3.2.2. Long-Term Outcomes and Ultimate Goals

The TOC describes three long-term outcomes: (1) increased FV purchases and intake, (2) improved food security, and (3) expanded economic benefits for participants, farmers, and food retailers. These allow GusNIP to contribute to the following ultimate goals identified by TOC contributors:Improve community health and economic well-being. The increase in FV purchases [17,50,54] leads to greater FV intake [17,18,19,20,21], which contributes to improved health outcomes [11,22,23]. Food retailers and farmers accrue economic benefits when participants buy FVs and other SNAP eligible items [11,53,55].Decrease health and wealth disparities. As participant health improves, health disparities in nutrition-related conditions may decrease. With better health, they and their families are more likely to succeed at school and work, thus increasing their earning potential and narrowing wealth disparities [56,57,58]. NIs supplement participant household income, increasing economic well-being.Sustain and expand a diverse and just local/regional food system. Local farmers, including farmers of color, benefit by selling produce to food retailers or directly to participants at farm direct sites, such as farmers markets [11,27,49,59,60,61,62,63].

#### 3.2.3. Assumptions

The TOC also includes assumptions that describe conditions and resources that exist outside of the program and are needed for program success (Table 3).

#### 3.2.4. Environmental Context

GusNIP exists in a complex environmental context. It operates within the larger food system and is one of many initiatives aimed at improving nutrition and reducing food insecurity. Some elements of this environmental context share GusNIP goals. For example, food system, food security, and nutrition organizations implement policies and programs to increase FV access and intake and food security among households with low incomes (e.g., The Special Supplemental Nutrition Program for Women, Infants, and Children (WIC) [67], Specialty Crop Block Grant Program [68], Pandemic EBT [69]).

Other aspects of the environment create obstacles and barriers. For example, issuance of NIs is limited to people participating in SNAP, thus excluding some non-citizens and other people in-need who do not meet eligibility criteria. Some potential participants may hesitate to enroll in SNAP due to associated stigma or concerns about immigration status [49]. Additionally, Congressional funding of GusNIP allows provision of NIs to only a small fraction of people receiving SNAP benefits, thus limiting program reach and impact [70]. The program would need approximately USD 5 billion annually to provide each of the 22 million households participating in SNAP with USD 20 worth of NIs per month.

GusNIP also competes with many influences on food purchasing choices. Consumer food choice is driven by taste preference, convenience, price, and availability [71]. Increasingly over time, ultra-processed foods have accounted for a growing share of food intake in the U.S. [72,73] and globally [74], thus potentially displacing FVs from the diet. Ultra-processed foods are inexpensive, widely available, convenient, hyperpalatable, and marketed heavily relative to FVs [75].

The environment varies across regions of the U.S., leading to differences in implementation across projects. For example, the status of the local/regional agriculture system determines the availability and affordability of local FVs in a community. Community characteristics such as rural versus urban location, number and types of retail food stores, infrastructure (e.g., transportation systems), and community food norms and values can influence where and how a project is implemented.

#### 3.2.5. The Future of GusNIP

This project focused on GusNIP as it currently operates. TOC contributors also described limitations and challenges of the present GusNIP program. They offered solutions about how to address them, largely focusing on increasing program reach and impact and enhancing equity (described in Table 4). TOC contributors noted that implementing solutions requires political will and increased program funding.

## 4. Discussion

The GusNIP TOC describes how and why GusNIP NI projects work. It is the product of an equity-centered, participatory, retrospective TOC development process. GusNIP provides funds and capacity-building support to grantees to implement projects that increase the purchase and intake of FVs among those who participate in SNAP, potentially improving their food security and providing economic benefits to participants, food retailers, and farmers. The program achieves this by providing NIs and increasing opportunities to purchase FVs, helping local food retailers provide FVs to participants in welcoming and accessible settings, and supporting local farmers to supply FVs at food retailers.

A TOC is a living, evolving theory. The ideas presented in Table 4 (GusNIP challenges and solutions) may be useful for further development of the GusNIP program and TOC.

The GusNIP program has made significant progress in providing access to FVs and supporting local economies. The Year Two GusNIP Impact report found NI participation is associated with higher FV intake. In 2020–2021, over USD 20 million in NIs were redeemed, bringing almost USD 40 million back into local communities [53]. Vericker et al. reported that retailers who participated in NI projects increased FV sales and store profits [12]. A 2021 economic modeling study estimated that for every dollar spent on incentives, up to three dollars in economic activity is generated [55].

Despite this success, GusNIP faces important challenges. Participant demand for incentives exceeds current funding levels, thus limiting access to GusNIP NIs to a small proportion of people participating in SNAP [70]. As described above (Section 3.2.4), an additional USD 5 billion may be needed to provide incentives to all people participating in SNAP. This represents a relatively small (5%) increase in current SNAP spending (USD 108 billion in 2021) [76]. There are efforts underway to increase GusNIP funding 4–12 fold in the 2023 farm bill [77]. Additionally, state and local governments are finding creative ways to fund NI expansion. Seattle is using dollars generated from its sugar-sweetened beverage tax and legislators in Rhode Island are advancing a bill to do the same [78,79]. Ultimately, integrating NIs into the SNAP program so that all participants automatically receive this benefit would be an efficient and cost effective approach [22,23,70,80]. A national program may also facilitate more extensive participation by larger retailer chains, where most SNAP benefits are currently redeemed [70], making it easier for participants to access and use NIs. Several local projects are already scaling their programs by working with larger retailers. For example, Washington State is working with Safeway (179 locations across the state), Reinvestment Partners works with Food Lion (510 locations across North Carolina), and Iowa has partnered with Fareway and Hy-Vee grocery chains throughout the state.

As GusNIP grows, equity impacts, particularly at the local level, will need to be carefully weighed. For example, including more large retailers may positively impact equity with regard to expanding access for participants and employment in the relatively diverse grocery sector workforce [81]. However, it will be important to continue to offer NIs at farm direct and smaller brick and mortar retailers that may be preferred by some participants, are located in higher need communities [82], and that include small businesses that are owned by people of color. Additionally, purchases at farm direct sites may make larger contributions to local economies compared to purchases at larger retailers [55]. Supporting diversity in types of retailers is a strength of the GusNIP program and allows sites to design projects that respond to their communities’ needs and priorities [11].

The administrative burden of the application process and grant management can be a challenge for grantees [12,70]. Grantees go through a competitive grant process, must match federal funds dollar-for-dollar with local resources, and are allowed a relatively low administrative indirect rate (federally negotiated or set at 10%) [14]. This may be a barrier to participation by higher-need communities that lack resources to prepare successful grant applications, cannot meet local match requirements, and are not able to cover administrative costs within the capped indirect amount. Prioritizing grant awards to lower-resourced organizations or higher-need communities could address this inequity.

Participation barriers exist for both retailers and participants. Many food retailers (particularly smaller and farm direct retailers) lack appropriate technology, including point-of-sale systems and electronic benefits transfer (EBT) equipment to efficiently participate [11,12]. Technical assistance and funding support should be provided to address these needs.

Awareness of GusNIP is low among people who participate in SNAP. In focus groups conducted among people participating in SNAP in North Carolina and Massachusetts, many were unfamiliar with NIs and wished there was more outreach [83,84]. Only 31% of those living near a FINI (the predecessor to GusNIP) participating retailer were aware of the program [85]. Our TOC focus group participants highlighted the importance of engaging the community in designing NI promotion campaigns and suggested including digital advertising through social media and apps. Digital advertising is a promising and scalable approach given the near ubiquity of smart phone ownership and social media usage across race and income levels [86,87,88]. Promotion strategies should be culturally tailored to each community and meet specific language needs.

Another barrier to participation in GusNIP is difficulty accessing NI retailers due to lack of participant transportation options or limited hours of retailer operation [44,56]. This barrier can be addressed by including community members in local project planning.

The use of TOCs for public health interventions is growing [30]. However, among TOCs described in peer-reviewed publications, few focus on nutrition programs in the U.S. and none on NIs specifically. A systematic review of public health intervention TOCs found only one related to nutrition, which focused on food security [30]. A scoping review of TOCs related to food security found that 90% were based in Asia and Africa and none addressed NIs [89]. We identified one additional peer-reviewed publication describing a nutrition-related TOC for a school food program [90]. While other nutrition-related TOCs appear in the grey literature, none focused on NIs in the U.S. [91,92,93].

### 4.1. Lessons Learned 

Several lessons emerged as we created this TOC. Authentic community engagement is a prerequisite for an equitable TOC development process. Community engagement added the voices of participants and practitioners, who contributed valuable insights, such as broadening the TOC to accommodate the local culture, values, and needs of each community. In addition to focus groups, workshops, and interviews, other strategies to deepen engagement could be considered: on-site community observations, listening sessions about NIs, community TOC development workshops that parallel those conducted with program professional partners and practitioners, and community surveys to reach a broader group of participants or practitioners. Authentic community engagement takes time and resources and should be planned for accordingly. It is critical that partners representing the different sectors participating in a program work collaboratively to build a TOC so they can learn from diverse perspectives. Broad participation yields a more robust TOC that is acceptable to all contributors.

The retrospective development of the TOC allowed for program implementation experience to shape it, a valuable addition to theory and academic evidence. However, TOC contributors frequently thought about how GusNIP could be improved beyond its current form, which was outside the TOC scope. It was important to validate and capture this more aspirational thinking for future program development while staying grounded in the present.

Adapting the traditionally in-person TOC creation process to a virtual setting suggested alternative approaches for developing a TOC. Eliminating in-person meetings reduced time commitment and expenses, allowing for participation by more diverse partners and participants. However, the typical TOC backwards mapping approach proved difficult to conduct in a virtual group setting, leading us to use interviews with individuals and small groups to generate input. A combination of in-person and virtual participation venues might be considered for future TOC development.

Balancing participation and efficiency merits careful consideration. For example, gathering preliminary information through program document and evidence reviews and interviews allowed construction of an initial TOC that provided a well-developed starting point for workshop discussions. There is no single TOC template or model. While our approach was based on published best practices [26,30], ultimately a format and structure specific to the GusNIP program and the TOC contributors was needed.

### 4.2. Strengths and Limitations

Our approach had multiple strengths. We developed a simple TOC designed for people without in-depth knowledge of GusNIP to make it useful for explaining the program to a broad audience rather than a more elaborate one aimed at an academic audience. We also developed a flexible TOC that could be adapted to local projects and community needs. This was particularly important for the GusNIP program given its support for local decision-making and variation across projects.

The TOC development process was robust. It was guided by current evidence on TOC methodology and integrated multiple sources of information (e.g., document and evidence review). Several engagement mechanisms were implemented to generate ideas and obtain feedback. TOC contributor participation was consistent during the entire process, in part due to trusted relationships between the NTAE, national partners and grantees, as well as between the project teams and the communities they serve.

The facilitators and the NTAE had subject matter expertise that helped in understanding the complex nuances of GusNIP. The facilitators had TOC development knowledge and served as an “objective broker of power relationships” [94] as the TOC was developed. Facilitators used strategies to manage contributor power dynamics during the workshops and provided opportunities for all TOC contributors to speak during workshop sessions. They increased the inclusivity of the process by offering multiple communication channels (e.g., virtual chat features, asking for input over email), allowing contributors to engage in ways that best suited them.

Our GusNIP TOC has several limitations. It is specifically focused on the GusNIP NI program and may not be generalizable to other NI programs. The food system is complex and varies across local contexts and it was not possible to include detailed consideration of every aspect of the food system environment that impacts GusNIP.

Capturing the full set of project implementation activities was not possible due to the numerous and diverse approaches used by grantees. The information about activities we found from GusNIP grantee reporting documents, while verified with the NTAE and validated with partners and practitioners, was incomplete.

We did not find evidence for every TOC component that was identified by TOC contributors. There was insufficient information showing how GusNIP contributes to realizing a diverse and just food system [11]. The pathway resulting in farmers supplying FVs to food retailers lacked evidence, as did the short-term outcome that food retailers know about NIs and value participants. For these TOC elements, we relied on the reported experiences of the TOC contributors.

It is possible that those who chose to participate in the TOC process may have been more supportive of GusNIP than other participants, partners, and practitioners. The participant focus groups were limited in number and representation of the geographic and cultural diversity of GusNIP projects. Logistical constraints did not allow for deeper participation or a decision-making role for participants. Finally, while reporting on program implementation and outcome indicators is recommended [30], these were outside the scope of this project.

## 5. Conclusions

The GusNIP TOC describes program goals, outcomes, activities, and the primary pathways connecting them. The TOC was developed using a retrospective, participatory approach centered in equity. It was designed to provide a shared understanding of how the program works for GusNIP national partners, grantees, practitioners, others implementing NI programs, evaluators, policy makers, and funders. The TOC provides a framework to foster collaboration across the GusNIP network, support program and project development, inform strategic planning, develop NI logic models, strengthen evaluation, and ultimately, improve and expand GusNIP so that it can continue to increase access to FVs for people participating in SNAP.

## Figures and Tables

**Figure 1 nutrients-14-02018-f001:**
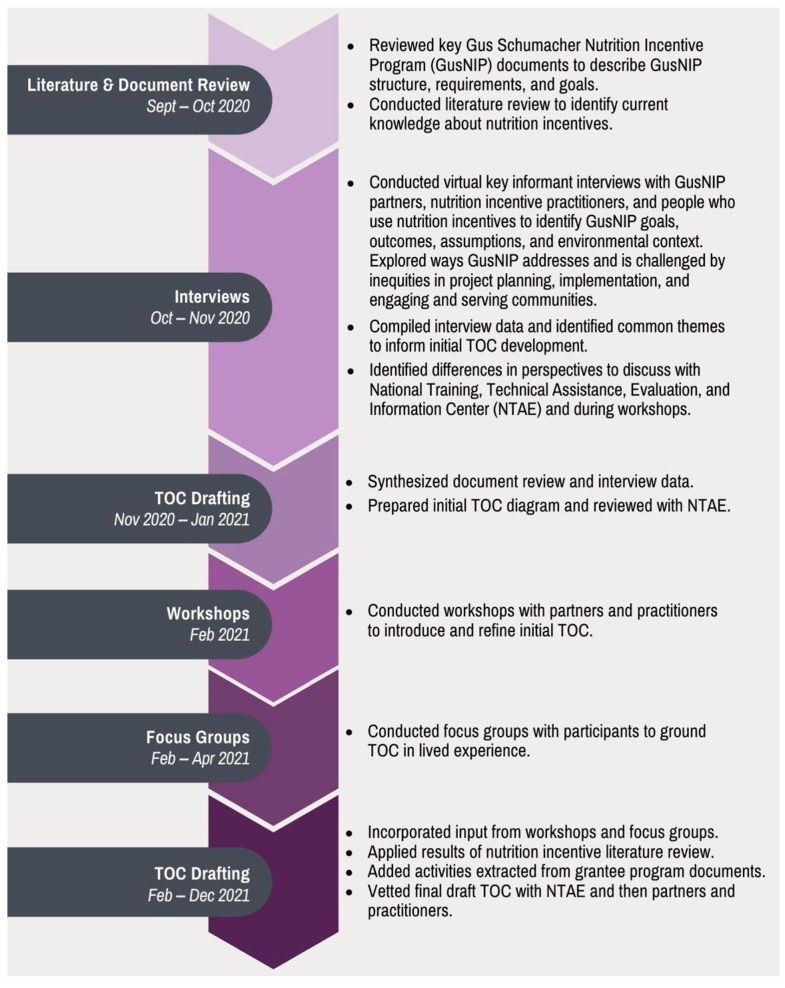
Theory of Change (TOC) Development Process and Timeline.

**Figure 2 nutrients-14-02018-f002:**
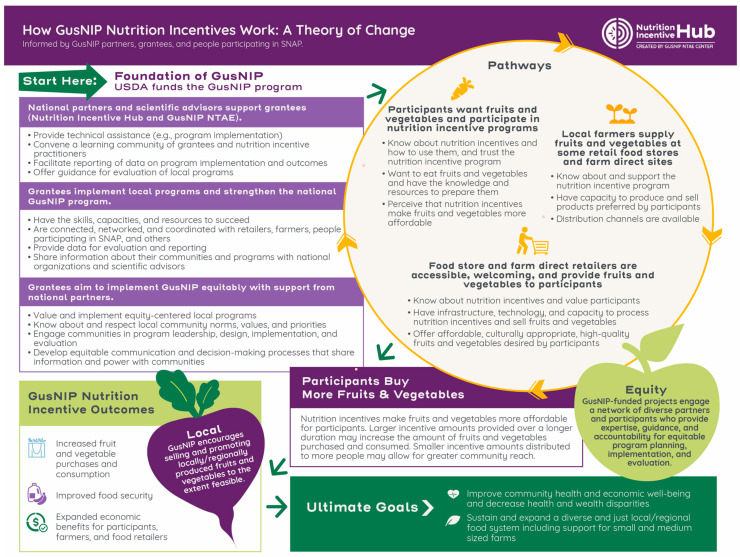
Gus Schumacher Nutrition Incentive Program Theory of Change Diagram.

**Table 1 nutrients-14-02018-t001:** Theory of Change (TOC) Contributors, Roles, and Engagement Method.

Type of Contributor (*n*)	Role	Interviewees ^a^ *n* = 26	Workshop Attendees ^b^ *n* = 25	Focus Group Participants ^c^ *n* = 17
Gus Schumacher Nutrition Incentive Program (GusNIP) Training, Technical Assistance, Evaluation, and Information Center (NTAE) (4)	Collaborated with facilitators to develop the TOC. Brought partners and practitioners to the TOC process. Contributed deep knowledge about the history of GusNIP, how the program works, and the operation of the NTAE and Nutrition Incentive Hub.	1	4	0
Partners (20)	Nutrition Incentive Hub partners and additional expert advisors from agriculture, food retail, academic, anti-hunger, and nutrition sectors. Brought expertise in nutrition incentive project impacts and best practices.	14	13	0
Practitioners (9)	GusNIP and other nutrition incentive project practitioners with experience in project implementation. Brought extensive knowledge about how nutrition incentive projects work.	9	8	0
Participants (19)	Community members who have utilized nutrition incentives. Brought lived experience of using nutrition incentives.	2	0	17

^a^ Key informant interviews were conducted to identify core GusNIP TOC elements and explore ways GusNIP supports and challenges equity. Eighteen interviews (1–3 contributors participated per interview), representing 18 different organizations and two participants took place. ^b^ Three workshops (20–24 contributors joined each workshop) occurred to develop a common understanding of TOC, introduce and refine the initial TOC, and ensure an equity lens contributed to TOC development. ^c^ Three focus groups (4–7 participants joined each focus group) took place to understand the participant perspective of how nutrition incentive projects work. Participants included 13 English speakers from Tennessee (*n* = 6) and Pennsylvania (*n* = 7) and 4 Spanish speakers from California.

**Table 2 nutrients-14-02018-t002:** Theory of Change Elements and Definitions.

Element	Definition
Activities	What a program does on a day-to-day basis to bring about outcomes. Activities are under the program’s control [27,31].
Assumptions	External conditions and resources that are needed for program success. Assumptions already exist, are not expected to be problematic, and are not within a program’s control. If the assumptions are not present, the program may not succeed, or unintended consequences may occur [31].
Pathways	Evidence- or experience-based logical and sequential connections between activities and shorter-term and longer-term outcomes [26,31].
Environmental context	The broader context in which the program operates. Includes community, cultural, structural, legislative, social, economic, environmental, and political forces that may shape the program. The program cannot control the environmental context [28].
Long-Term Outcomes	What a program wants to achieve—the purpose of the program [31].
Short-Term Outcomes	“A state or condition that must exist” [31] for the program to achieve the long-term outcomes. An outcome represents a change in knowledge, skills, attitudes, and behaviors in people or changes in processes or systems of an organization [27,31].
Ultimate Goal(s)	The visionary “big picture” change the program contributes to but is beyond what the program can achieve on its own [31].
Theory of Change (TOC)	A living, theoretical model that explains how and why change is expected to happen. A TOC is visually represented in a TOC diagram and described in an accompanying narrative. It is developed through a participatory process that includes multiple and diverse perspectives [26,31].

**Table 3 nutrients-14-02018-t003:** Pathways and Corresponding Assumptions.

Pathway	Assumptions
Foundation of GusNIP supports local projects	National partners and scientific advisors are willing to engage in GusNIP and funds are available to support them. Local organizations with the capacity to implement GusNIP programs exist and participate. Grantees and national partners value equity.
Participants want fruits and vegetables and use nutrition incentives to buy them	Participants want to eat more fruits and vegetables. Participants find fruits and vegetables expensive, creating a barrier to fruit and vegetables purchases and intake [5,64]. Participants value and want to support the local food system [56,65,66].
Food retailers participate in projects	Food retailers want to participate in the program. Food retailers are, or are willing to become, authorized Supplemental Nutrition Assistance Program (SNAP) retailers.
Local farmers provide fruits and vegetables to projects	In some communities, farmers supply fruits and vegetables desired by participants. In some communities, farm-to-food retailer distribution channels are available.

**Table 4 nutrients-14-02018-t004:** GusNIP Challenges and Solutions.

Participant Experience	
Challenges	Solutions
Nutrition incentives (NIs) only reach a small portion of people participating in the Supplemental Nutrition Assistance Program (SNAP).	Expand program funding so everyone who participates in SNAP can receive NIs.
People with low incomes who are ineligible for SNAP cannot receive NIs.	Expand program eligibility beyond SNAP income eligibility threshold.
The amount of NIs provided is not adequate for all participants to meet U.S. Dietary Guidelines for Americans fruit and vegetable (FV) recommendations.	Provide a minimum monthly benefit and/or eliminate the participant match needed to earn NIs.
NIs are limited to FVs and do not address the full dietary needs of all participants.	Allow purchases of other healthy foods with NIs in addition to FVs.
Some participants cannot access participating food retailers due to transportation challenges and hours of operation.	Identify and recruit retailers that are easy for participants to access.
Some participants do not feel welcomed at participating food retailers.	Increase participation by culturally diverse retailers and farmers and provide trainings to all retailers on creating welcoming, culturally tailored environments.
**Program design and implementation**	
**Challenges**	**Solutions**
Not all people using SNAP who are eligible to receive NIs know about them.	Engage community to identify and implement effective NI promotion and outreach strategies.
Food retailers (particularly smaller retailers) lack appropriate technology for issuing and redeeming NIs, including point-of-sale systems and electronic benefits transfer equipment.	Support implementation of effective issuance and redemption processes and technologies that meet food retailer and participant needs (e.g., point of sale technology that decreases burden on food retailer and decreases stigma for participants at point of sale).
The process for an organization to apply for GusNIP funding is challenging, especially for lower-resourced organizations.	Provide support and technical assistance to new and lower-resourced applicants.
The administrative burden of program management, including local funds to meet federal grant match requirements, can be challenging, especially for new and lower-resourced projects.	Address funding needs and capacities of organizations (especially minority-led or under-resourced organizations) for project administration, meeting local match requirements, capacity building, and evaluation.
Local projects may not fully understand or meet the needs of the community served.	Expand community member role in project design and implementation and compensate them for their involvement.
Interested partners (e.g., food retailer, food security, nutrition, local/regional food system sectors) are siloed.	Foster communication and partnership across interested partners.

## Data Availability

The data presented in this study are available upon request to the corresponding author. The data are not publicly available since confidentiality was promised to workshop, focus group, and interview participants.

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
