# Peer review of "How Does the Gus Schumacher Nutrition Incentive Program Work? A Theory of Change"

_nutrients, 2022, doi:10.3390/nu14102018_

Round 1
Reviewer 1 Report
How does the Gus Schumacher Nutrition Incentive Program 2 Work? A Theory of Change
- This article provides a Theory of Change (TOC) for GusNIP. Strengths include nice readability, clear organization, and interesting source material from focus groups and interviews with a range of stakeholders.
- My leading suggestion, for the authors’ consideration, is a more critical or independent empirical approach. In tone and content, the approach of this manuscript seems determinedly supportive rather than empirically curious about areas of strength and weakness for GusNIP, which could be a missed opportunity. A critical inspection of the program could (a) be more persuasive with readers by presentin the authors as more independent of the material and (b) more beneficial for the program by more pointedly noting areas to work on.
- The article and TOC could give more attention to cost and efficiency. For this readership, economic jargon is not advantageous, but the basic point could be addressed using terms such as cost, efficiency, good governance or good program management principles, or good value for money. Without these themes, the favorable upbeat approach to GusNIP seems predetermined, because the program could not possibly be harmful, and it has a wonderful feel-good vibe supporting participants and small farmers, so without attention to cost per person served, efficiency, and scalability, there really was no possibility of empirically finding anything critical.
- A leading question is whether the emphasis on farmers’ markets and CSAs in GusNIP serves the interest of low-income consumers well. At one point, participants mentioned high cost of FVs in farmers’ markets as a concern, but the authors did not wrestle with the implication of this. I think it likely that there was, or certainly would be with probing, substantial primary source material from the participants in the focus groups and interviews suggesting some skepticism of farmers’ markets as an easy access source and some preference for other retail formats. Brick and mortar stores have some small role in GusNIP, but not the major retailers where participants get most of their food, which has strong negative implications for scalability in the Theory of Change.
- Related, the article discusses equity as a priority, but overlooks some of the biggest equity concerns. The commercial U.S. food retail sector is a major employer for low-income Americans and people of color, but farmers’ market employment is notably disproportionately non-Hispanic white. The manuscript notes some exceptions, including opportunities for farmers of color, but I think a self-critical empirical approach would have led the authors to note this issue more starkly. If GusNIP is to make a big difference in employment as well as FV intake, the emphasis on local food systems might need to be critically considered.
- The article sends mixed signals about whether local farmers’ markets and CSAs are key political stakeholders in the Theory of Change. The ultimate goals at lines 214-216 are focused on the FV needs of participants rather than producers. Later in Figure 2, I think farmers and local food system suppliers are indicated as key stakeholders.
- The TOC could be hard-headed about prospects for growth by indicating what percentage of SNAP benefits currently are spent in farmers’ markets and local food outlets. A Theory of Change must have either some theory of staying small permanently or of scaling up, and if there is scaling up, how so.
- Optionally, Sara John et al. in American Journal of Public Health on nutrition incentives may be a relevant source.
Reviewer 2 Report
I really found reading the manuscript interesting. Well designed methods and the results are presented properly. Have a few minor comments to improve the manuscript.
Line 46: delete "projects"
Line 92: 4-7 participants...based on what? is that the idea numbers? Ref??
Line 116: Reason for selecting these two search engines
Lines 116-117: Better to describe (2-3 sentences) how articles were searched
Line 153: Notes all non-verbal communication or cues?
Lines 166-169: Prefer to move this section to the first of methods, because this took place first before kicking off the study.
Table 4: Heading (Font- consistency)
Table 4 :TOC: from Participants- any challenges needed to address for the future?
Thank you.
Round 2
Reviewer 1 Report
The revision is responsive.